# Surface Water Microplastics in the St. Lawrence River and Estuary in Canada

**Valerie S. Langlois** [1]*, **Tuan Anh To**[1], **Eve Larocque**[1], **Julien Gigault**[2], **Raphael A. Lavoie**[3]

**1** Institut national de la recherche scientifique (INRS), Centre Eau Terre Environnement, Quebec City, QC, Canada, **2** Takuvik Laboratory, IRL3376 Centre National De La Recherche Scientifique (CNRS) -Université Laval, Quebec City, QC, Canada, **3** Environment and Climate Change Canada (ECCC), Science and Technology Branch, Quebec City, QC, Canada

* valerie.langlois@inrs.ca

**Data availability statement:** All relevant data are within the manuscript and its Supporting

## Abstract

Microplastics (MPs) are synthetic or semisynthetic polymers that are widely distributed throughout most ecosystems and have the potential to be harmful to living organisms. In this study, we assessed the MP fraction in the top 40 cm of surface water in response to varying salinity levels at 11 distinct sites across the St. Lawrence River and Estuary (SLRE). We employed two sampling nets of different mesh size to collect MPs (100 and 300 µm). These nets were simultaneously towed in parallel from a vessel during three separate sampling events at each designated site. Filtrates collected from these samples underwent analysis of plastic fibers, fragments and spheres utilizing Fourier Transform Infrared Spectroscopy (FTIR). Data unequivocally confirmed the presence of MPs at 100% of the sites sampled within the SLRE. The most abundant categories of MPs identified were the fibers, followed by fragments and spheres. The FTIR analysis revealed the predominant materials to be polyester, polyethylene, polypropylene, nylon, and polystyrene. Notably the findings also suggest MPs are more likely aggregating when salinity increases. This work offers valuable insights into the distribution and behavior of MPs contributing to the preservation and management of water resources.

## Introduction

Microplastics (MPs; size ranging from 5 mm to 1 µm) are recognized as the breakdown products of macroplastics, such as tire abrasion particles or textile washing fibers [1]. Consequently, the substantial proportion of marine macroplastics are anticipated to degrade over time, leading to a putative increase in MPs concentrations in the coming years. Indeed, MPs have been detected across various environmental compartments, including soil [2], sediment [3], clouds [4], wildlife [5], human lungs [6], and most notably, in aquatic environments. An estimated 109 million metric tons (Mt) of plastics have accumulated in rivers and 30 Mt in oceans, comprised of 88% macroplastics and 12% microplastics [7].

The adverse effects of MP exposure on the health of living organisms' have been extensively documented (reviewed in [8–10]). In humans, exposure to MPs has been linked to oxidative stress, metabolic disorders, immune response, and both reproductive and developmental toxicity [9,11]. Fish exposed to MPs have exhibited signs of neurotoxicity, growth retardation,

Information files. Data and R scripts are shared on an open access repository on at https://github.com/ECCC-lavoie-ecotox/MPs_St-Lawrence in a way that follows best practice and facilitates reproducibility and reuse.

**Funding:** The Canada Research Chairs and the Centre national de la recherche scientifique had no role in study design, data collection and analysis, decision to publish, or preparation of the manuscript

**Competing interests:** NO authors have competing interests

and behavioral abnormalities [8]. Additionally, microalgae living in MP-contaminated water have shown reduced nutrient availability [12]. The spectrum of reported health impacts of MPs is likely just the beginning, as global research efforts continue to investigate various species.

Among the array of environmental factors, salinity has emerged as a significant variable, capable of altering the electrostatic properties of MP surfaces [13] and nanoplastics [14]. Several studies have highlighted the relation between salinity and the sorption of environmental contaminants (antibiotics [15], metals [16], organic micropollutants [17]). Interestingly, Zhou et al. (2021) showed that MPs accumulate at the halocline in the sea, suggesting variable MP behavior with varying salinity levels [18].

The St. Lawrence River and Estuary (SLRE)'s hydrographic system drains approximately 25% of world's freshwater, with over 45 million people living within its vicinity [19]. Despite the immense size of this basin, few MP studies have been conducted in the SLRE [20–22], with most analyses focusing on higher density MPs that sediment at the bottom of the SLRE. However, water surface is known to carry approximately 75% of MPs with less than 1.1 g/cm³ of density [23]. The objectives of this study were two-fold, with the first being the quantification and characterization of MPs sampled at 11 sites along the SLRE using two different sized mesh nets (100 μm and 300 μm mesh). The second objective was to determine if a relationship exists between MP concentrations and shifting salinity within the SLRE. We hypothesized that urbanized sites and higher salinity waters will show profiles of high MP concentration.

## Materials and methods

### Sample collection and processing

Based on the salinity gradient from the SLRE, 11 sampling sites were sampled near the following cities: Varennes, Contrecœur, Sorel-Tracy, Trois-Rivières, Portneuf, Quebec, Montmagny, La Pocatière, Baie Saint-Paul, La Malbaie, and Trois-Pistoles (Fig 1). Sites were sampled from

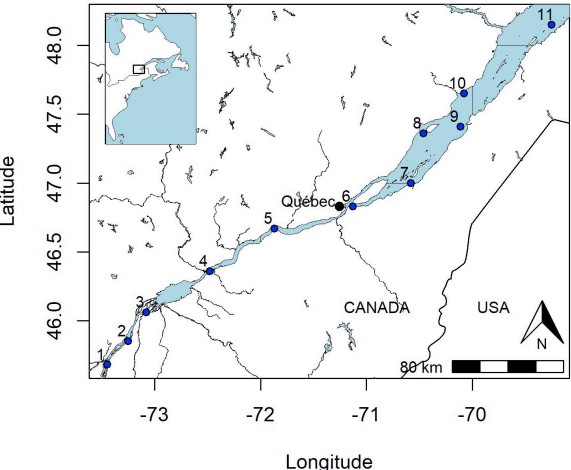

**Fig 1. Map of the St-Lawrence River and Estuary (SLRE) sampling sites.** The nearest cities are used for site names. Fluvial sites: 1 = Varennes, 2 = Contrecœur, 3 = Sorel-Tracy, 4 = Trois-Rivières, 5 = Portneuf, 6 = Québec. Estuarine sites: 7 = Montmagny, 8 = Baie-Saint Paul, 9 = La Pocatière, 10 = La Malbaie, 11 = Trois-Pistoles. Basemap source: Government of Canada; Natural Resources Canada (https://open.canada.ca/data/en/dataset/8ba2aa2a-7bb9-4448-b4d7-f164409fe056) and Natural Earth (public domain): http://www.naturalearthdata.com/. Contains information licensed under the Open Government License – Canada (https://open.canada.ca/en/open-government-licence-canada) and Natural Earth (https://www.naturalearthdata.com/about/terms-of-use/).

May 26 to June 19, 2021. At each sampling site, three transects of 20 min (approx. 200,000 L) were performed using two different sized mesh nets: 100 μm (Poly-Mer©, Québec, Canada) and 300 μm (Manta, Québec-Ocean, Quebec, Canada). The diameter of the nets was 33.2 X 15.3 cm for Poly-Mer© and 40 X 20 cm for Manta. Building on previous studies [24,25] that showed that MPs tend to concentrate in the first 15 cm of the water column, nets were positioned to filter the top 40 cm of water surface. A flow meter (KC Digital flow meter model 23.090) was used to estimate the volume that was filtered during each transect. GPS localisation measurements allowed sampling of three comparable transects per site. Three blanks are also made using distilled water at each sampling site to obtain a total of nine samples per site. Weather conditions were recorded as it is known there is MP concentrations increased in surface water following rain events [26]. No permits were required to sample water in the SLRE.

After sampling, all the matter collected on the filter was resuspended with 1 L of deionized (DI) water and preserved in closed glass jar at 4 °C until filtration. For field blanks, 1-L DI was poured into a glass jar on the boat and stored at 4 °C until filtration. Filtration was done using the Whatman Ashless Quantitative filters of 20–25 μm x 110 mm. Filter samples were stored at -20 °C in Petri glass dishes covered with aluminum foil until sample digestion.

## Digestion process

Due to the presence of abundant plant material on filters, several digestion methods were tested, including the digestion with hydrogen peroxide ($H_2O_2$) at 30% (v/v; Thermo Fisher) at different times and temperatures: 6 h at 65 °C, 24 h at 50 °C, 24 h at 25 °C and 4 h at 50 °C with Fe(II). At last, the $H_2O_2$ + Fe was the most suitable for the samples. The material collected on the filter was transferred carefully into a clean Petri glass dish. During this step, the filter was rinsed with 10 mL of $H_2O_2$ at 30% to collect most of the material available then 10 mL of catalyzer was added (FeSO$_4$ 0.05 M, $H_2SO_4$ 0.5M, Fisher) [27,28]. The digestion was performed at 50 °C [29] for 4 h and 10 mL of $H_2O_2$ 30% was added every hour. Once digested, the mixture was filtered using a 90 mm glass fiber grade F with 0.7 μm pore size (Whatman). The filter membrane was pre-marked with 20 x 20 mm grid to simplify the counting. Samples were air dried and preserved at 4 °C until the counting step. All tools and material used for the entire process were made of cotton, metal, glass or ceramic to avoid plastic contamination.

## ATR-FTIR analysis

Potential MPs were pre-selected following Hidalgo-Ruz et al. (2012)'s guidelines under a binocular microscope (Wild M10 Leica) with 100X magnification [30]. The particles were examined with an Attenuated Total Reflection-Fourier-Transform Infrared Spectroscopy (ATR-FTIR; Nicolet iS50, Thermo Fisher) for 16 scans at wavelengths ranging from 4,000 cm$^{-1}$–500 cm$^{-1}$. The infrared spectroscopy (IR) spectra were collected and compared against a library by Thermo Fisher (Aldrich polymer, Hummel polymer, Synthetic fibers, Additives) by OMNIC™ Specta Software. The MPs that scored more than 70% similarity to the library spectra were annotated with their respective material [31], and were then classified according to their polymer name (e.g., polyethylene) and shape (e.g., fibers, fragment and sphere).

## Data analysis

The number of particles within each towing were standardized to the volume filtered into particles of plastics per million of liters (PPML) using the following equation (1):

$$\frac{\left(Total\ of\ particles\ found\ on\ a\ half\ of\ the\ filter\right) \times 2 \times 10^6}{Volume\ of\ water\ filtered\ \left(L\right)} \tag{1}$$

Note that the second half of the filter was kept for future investigations of nanoplastics. Replicates at a given site were considered independent samples and were treated separately in models. Seral competing linear models were run using a combination of area (fluvial: Montreal to Québec vs. estuarine: downstream of Québec), sampling sites and mesh net sizes (and their interaction) on overall sum of polymers and on sums per shape of plastics. Site was nested within area. The most parsimonious model was selected by Akaike Information Criterion on small sample size (AICc) using the package *MuMIn* [32]. When the difference in AICc (ΔAICc) between the best model and the second-best model was high (>2), and when Akaike weight (w; model probability) of the top model was high (>0.9) [33] the output for the model with the lowest AIC was the most likely for the dataset. Parametric models were run after conditions of normality (Shapiro test) and homogeneity of variances (Levene's test) on residuals were respected. Values were log-transformed (base 10) when necessary to respect assumptions. Multiple comparison tests were run among levels of factors using the *multcomp* package [34].

To determine diversity and assemblage of structure of microplastics among sites, we used indicators of alpha diversity (diversity within a sample) using Shannon and Simpson diversity index and beta diversity (dissimilarity between samples) using the Bray-Curtis dissimilarity. The assemblage structure of microplastics was measured using a non-metric dimensional scaling analysis (nMDS) with the Bray-Curtis dissimilarity index in the function "metaMDS" from the *vegan* package in R [35]. Mesh size was used to make two-dimensional ordinations. We tested for differences between the assemblage structures of microplastic between the fluvial (Montreal to Québec) and estuarine (downstream of Québec) portions of the St. Lawrence River by mesh size among matrices by running PERMANOVAs using the function "adonis2" in the vegan package [35]. The number of permutations was set to 999. The function "permtest" was used to test that the dispersions among groups were equal. Differences were considered significant when $p < 0.05$. All analyses were conducted in R version 3.6.2 [36]. Data and R scripts are shared on an open access repository on at https://github.com/ECCC-lavoie-ecotox/MPs_St-Lawrence in a way that follows best practice and facilitates reproducibility and reuse.

## Results

Fibers and fragments were the most common particle shapes found at all sites and the order of importance changed among sites and between mesh sizes (Figs 2 and 3). Approximatively, 20–85% were fibers, 10–70% were fragments and 0–20% were spheres. Overall, more MPs were captured using the finer mesh size. Polyester, polyethylene, and polypropylene were the polymer types that were the most abundant in both mesh sizes at most site and accounted for over 85% of the total amount of particles overall (Figs 2 and 4). Many particles of nylon were found in one replicate at Portneuf accounting for 60% of all particles.

Several models were built to explain variation in the sum concentration of particles and included a combination of site, area (fluvial vs estuary) and mesh size. Three models were equally parsimonious based on AICc (ΔAICc < 2): area, the null model and mesh size (Supplementary Table S1). This suggests that the sum of MP particles is randomly distributed among sites and mesh size. This indicates that MPs are ubiquitous in the St. Lawrence and that point sources are not as important as hypothesized to explain MP abundance. Concentrations of MPs were not different between areas (t = 1.6; $p = 0.12$), but were different among sites ($F_{(10,50)} = 2.3$; $p = 0.025$; Fig 3), although the coefficient of determination was low ($R^2 = 0.32$). Mesh size remained non-significant ($F(1,59) = 0.8$; $p = 0.38$). MP concentrations (mean +/- SD) for all 11 sites combined was 72.3 ± 93.0 PPML and was highest at the La Pocatière site (111.0 ± 120.0 PPML) and lowest at the Sorel site (38.7 ± 40.8 PPML). A subset using fiber

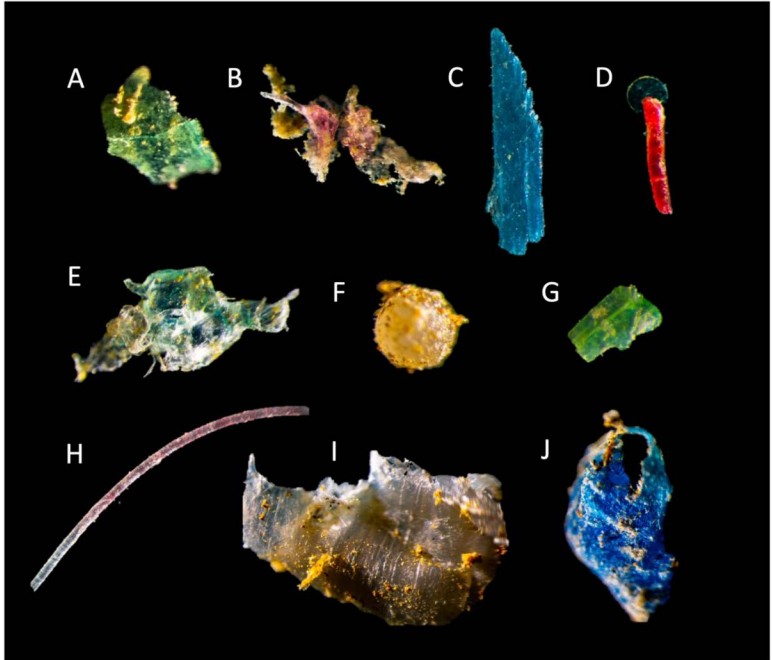

**Fig 2. Microphotographs of microplastics found in the St. Lawrence River. (A)** Green polyethylene fragment, Portneuf, >300 μm **(B)** Beige polyethylene fragment, Portneuf, >300 μm **(C)** Blue polyolefin fragment, Portneuf, >300 μm **(D)** Red polyethylene fiber, Baie Saint-Paul, >300 μm **(E)** Green polyethylene fragment, Trois-Rivières, >100 μm **(F)** Transparent and yellow polystyrene bead, Baie Saint-Paul, >300 μm **(G)** Green polystyrene fragment, Sorel, >100 μm **(H)** Beige polypropylene fiber, Trois-Rivières, >100 μm **(I)** Transparent and yellow polyethylene fragment, Trois-Pistoles, >300 μm; and **(J)** Blue polyethylene fragment, Trois-Pistoles, >300 μm.

concentration showed that two models were equally parsimonious based on AICc (null and mesh size; Table S1), but none of the factors, including area and site were significant when taken separately (p > 0.05). This suggests that fiber particles are randomly distributed spatially, and that mesh size does not influence the abundance of particles sampled.

When fragment concentrations were tested separately, three models were equally parsimonious (area, site + are, site; Table S1). Concentrations of fragment particles were higher downstream compared to upstream (t = 2.6; p = 0.011) and sites were significantly different ($F_{(11,47)}$ = 3.0; p = 0.004; $R^2$ = 0.41; Fig 5; Table S1). Measures of Shannon and Simpson diversity of microplastic particles were not significantly different among sites or between sections of the SLRE (fluvial vs. estuarine; p > 0.05; Table S2). This suggests that the diversity of particles was homogenous among sites. The nMDS plots suggested that the assemblage structure of MPs was different between the fluvial (freshwater) and estuarine (brackish water) portions (Fig 6). This was confirmed by the PERMANOVA showing a statistical difference when both mesh size nets were combined ($F_{(1,60)}$ = 3.3; p = 0.016; Fig 6A) and the small mesh size (Poly-Mer; ($F_{(1,28)}$ = 2.4; p = 0.047; Fig 6C), but not for the large mesh size (Manta; $F_{(1,31)}$ = 1.9; p = 0.09; Fig 6B).

## Discussion

This project analyzed MP levels at 11 sites across the St. Lawrence River and Estuary (SLRE) and confirmed that MPs are ubiquitous throughout all sites. In general, regardless of the sampling points, it appears that the MPs most often found in oceans (i.e., polyethylene, polypropylene, and polystyrene) [37] are not the predominant ones measured in the SLRE. Indeed,

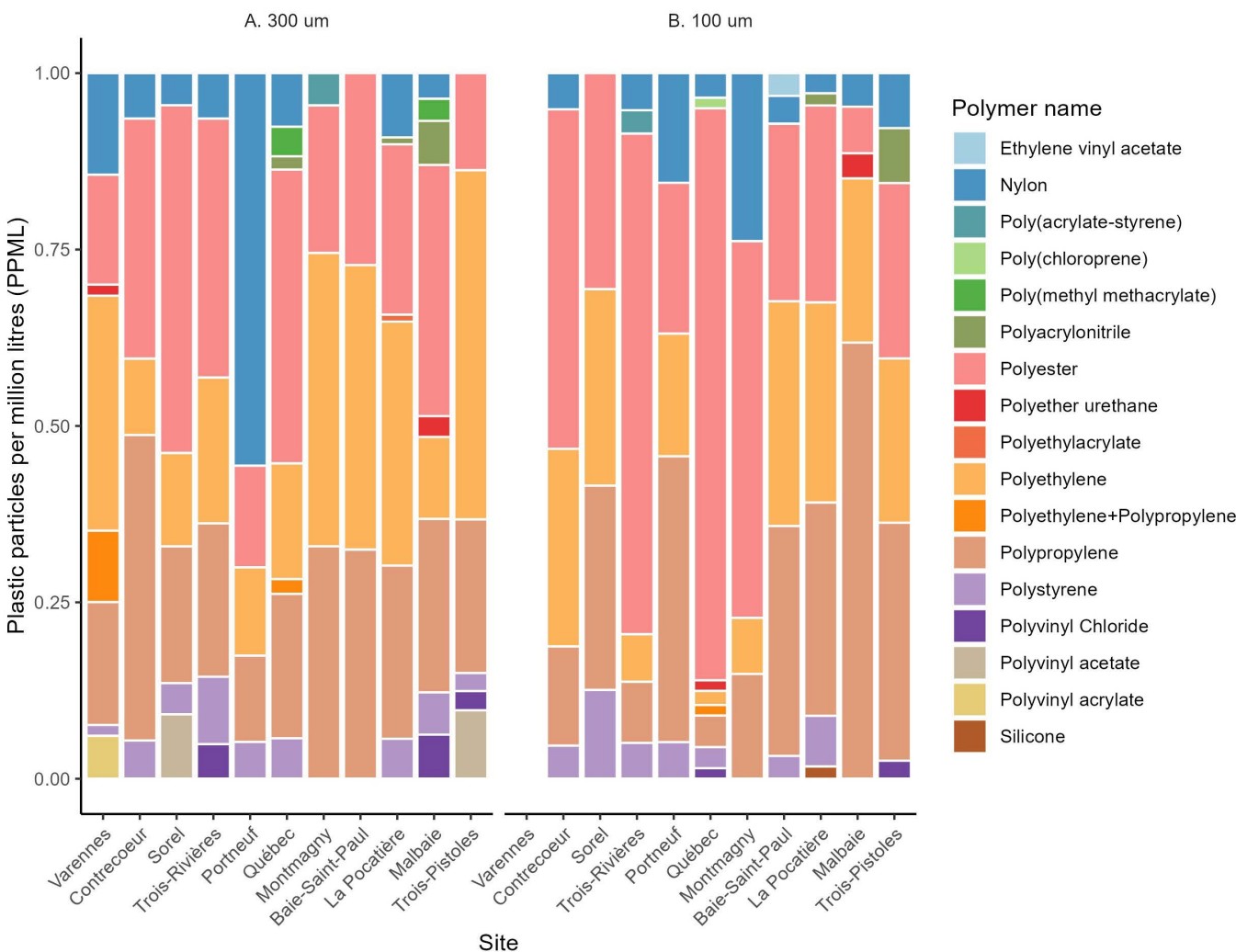

**Fig 3. Proportion of microplastics by shape (fiber, fragment and sphere) for A. 300 μm and B. 100 μm mesh size across all sites.** Left to right: upstream to downstream; sites 1 to 11.

MP sources in rivers are often closer to the points of discharge of less degraded plastic debris in the environment, such as urban, industrial and agricultural areas, where various types of plastics are being used and released into the environment. In contrast, in the ocean, MPs can originate from broader sources, including those discharged by rivers, as well as maritime activities, coastal dumps, etc. In rivers, a greater abundance of plastics associated with urban and agricultural activities can be found, while in the ocean, a greater variety of plastics linked to maritime activities can be found [38].

In the case of polypropylene and polyethylene, their high mechanical resistance and physical properties make them resistant plastics. This implies that their fragmentation occurs only under strong oxidative stress such as photooxidation, which is only prevalent in oxygenated areas of ocean exposed to ample sunlight such as the ocean surface. The large debris of polyethylene and polypropylene, even if they are in the minority among all plastics, will therefore transit through rivers and estuaries and end up in the ocean to fragments into MPs as previously modelled [39]. As for polystyrene, its quantity is relatively low compared to other plastics [39]. This can be explained by its more specific applications (e.g., construction,

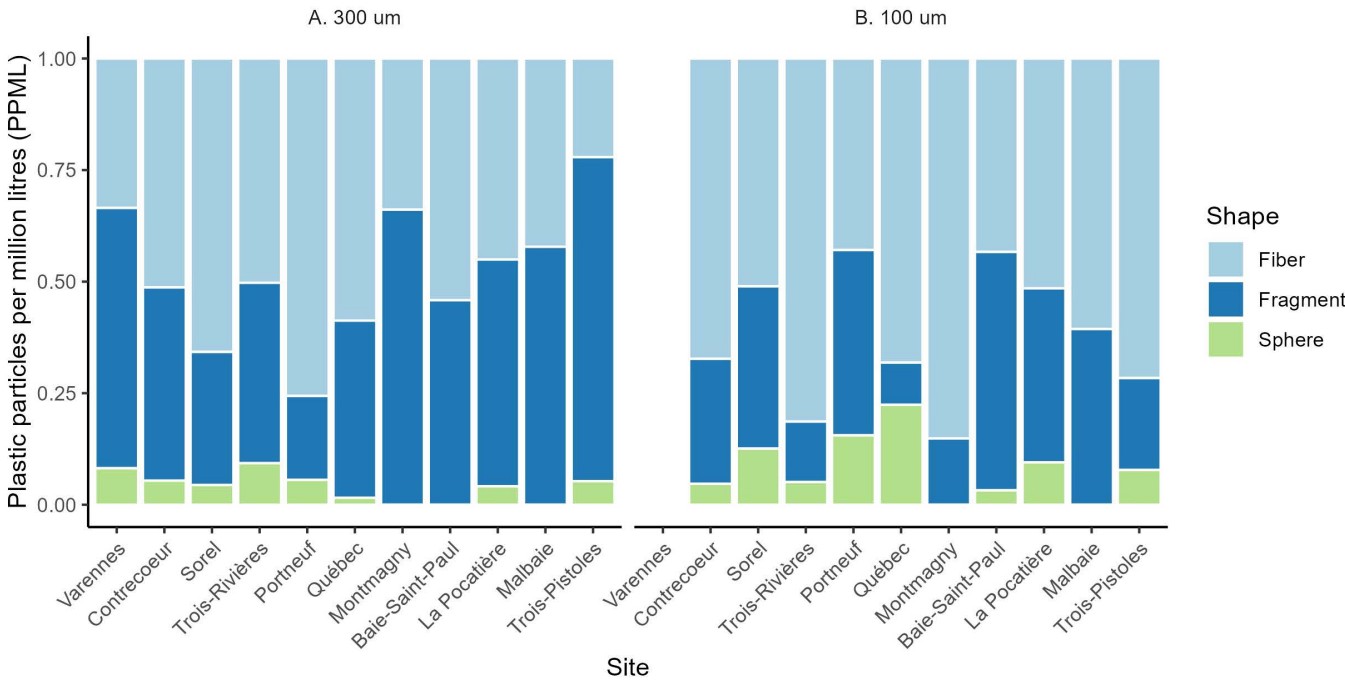

**Fig 4. Proportion of microplastics by polymer name for A. 300 μm and B. 100 μm mesh size across all sites.** Left to right: upstream to downstream; sites 1 to 11.

insulation, and packaging) and especially by the much more advanced recycling pathways as compared to other polymers.

Indeed, a recent study conducted on a Chinese river has demonstrated that the major sources of plastics in the river are primarily due to agricultural activities, domestic wastewater and industrial activities [40]. Among these activities, domestic wastewater mainly involves the discharge of washing machines, which release large quantities of micro- and nanofibers into the wastewater [41]. Indeed, this is confirmed with the analysis as a large proportion of polyester and nylon fibers were found across all sampling stations, particularly in proximity to the most anthropogenically influenced areas (i.e., Quebec City and Trois-Rivières City) as observed elsewhere [42]. It is worth noting that MPs that accumulate in wastewater treatments plants can be also reintroduced into the environment with sewage sludge during its inciner- ation or application as compost in agricultural application [43,44]. Zubris et al. (2005) have showed polyester fibers as effective indicators of sewage sludge contamination on agricultural fields [44]. Caution should also be considered when interpreting fiber MPs caught by the Manta as its underestimate particles smaller than its mesh netting [45].

We expected to find higher MP concentrations upstream than downstream considering the high population density living nearby the Great Lakes and SLRE from the United States and Canada, but lower concentrations were found than elsewhere. For example, Luo et al. (2019) measured concentrations around 2,000,000 PPML near major urban centers in China [46]. This difference can be explained by the hydrology and typology of the SLRE. Indeed, studies have shown that shoreline accumulation and deposition in the sediment were not considered as they can be collected and decontaminated [22]. Castañeda et al. (2014) have estimated MP loadings in SLRE sediment up to 3,800,000 PPML [22] at sites which have accumulated continuously for over 50 years compared to transitioning surface water [47]. In another study, fragments and fibers are the most prevalent particles found in the sediment [48] in

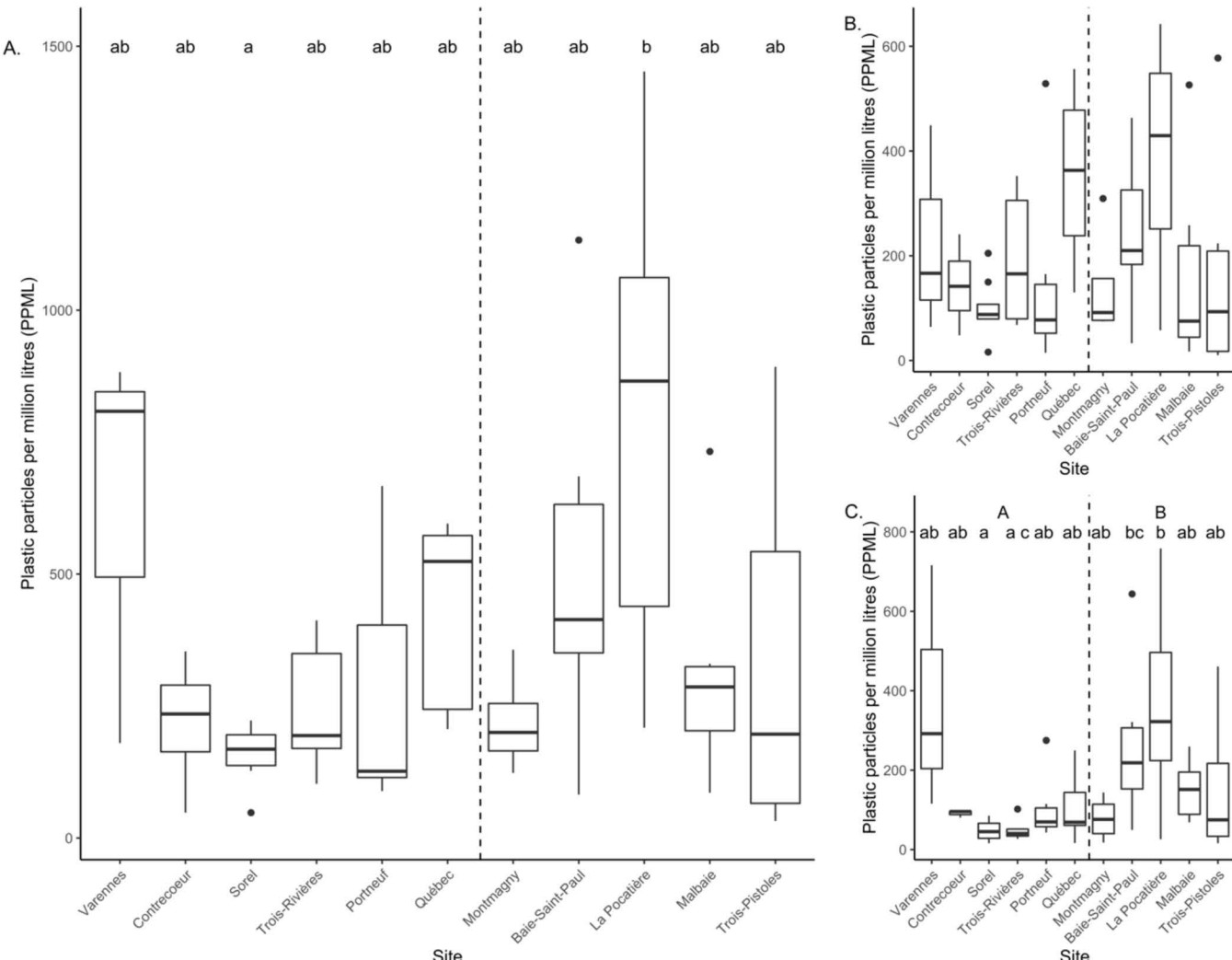

**Fig 5. Particles of plastics per million of liters (PPML) shown as A. total particles, B. fiber particles and C. fragment for each site.** Central lines in the boxes represent the median and the boxes show the interquartile range (25-75th percentile) and lines show the minimum maximum. Values were log$_{10}$-transformed for statistical analyses but presented in their raw form for visualization. Sites that share common lower-case letters do not vary statistically and the difference between areas is shown with upper-case letters. Sites are ranked from upstream to downstream (left to right; sites 1 to 11) and areas are separated by a vertical dashed line.

South Africa, which could correlate with the variation in shape observed from the sampling campaign.

Concentrations found in this study are consistent with estimates provided by various particle flux models to the oceans [38,41]. The most recent models even contradict previous assumptions by stating that it is the smallest rivers that represent the major sources of plastics in the oceans [49]. Finally, these data also fell within the range of concentrations previously reviewed by Koelmans et al. (2019) allowing for comparison of analytical approaches [50]. This indirectly confirms that both sampling systems used, and the analytical approach yielded acceptable results concerning fluxes and, most importantly, characterization and quantification methods [50].

However, the unexpected re-increase in the concentration from the river to the estuary is surprising. Indeed, due to the decreased population and the increased volume of water,

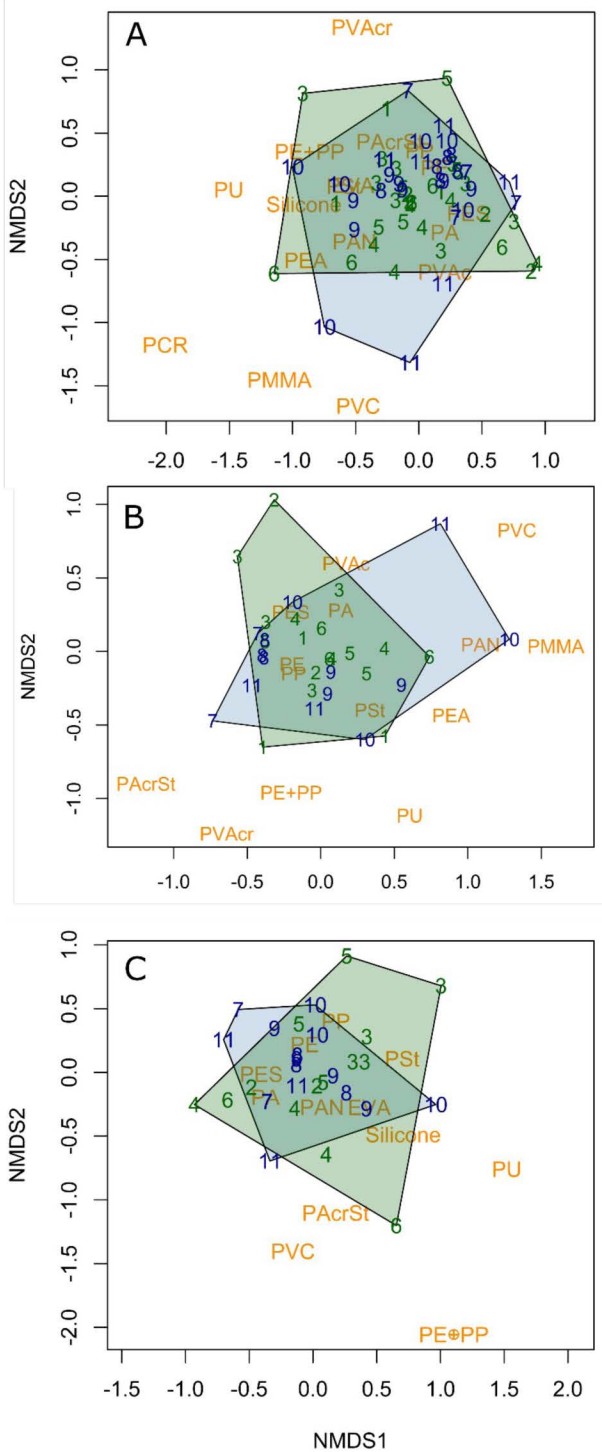

**Fig 6. nMDS assemblage structure of microplastics for A. both nets combined (stress value = 0.178), B. 300 μm mesh size net (Manta; stress value = 0.155) and C. 100 μm mesh size net (Poly-Mer; stress value = 0.185).** Polymer types are in orange font and sites are presented by numbers with fluvial sites (1 = Varennes, 2 = Contrecœur, 3 = Sorel-Tracy, 4 = Trois-Rivières, 5 = Portneuf, 6 = Québec; green font and polygon) and estuarine sites (7 = Mont-magny, 8 = Baie Saint-Paul, 9 = La Pocatière, 10 = La Malbaie, 11 = Trois-Pistoles; blue font and polygon). Dash line separates fluvial to estuary sites.

a decrease of the MP concentration was expected [46]. Numerous studies have highlighted a direct link between MP concentration and population density in downstream regions [41,51,52]. Wagner et al. (2019) showed that plastic concentrations remain stable in rural areas but exhibit a linear increase in urban areas due to the urban inputs [52]. On the opposite, Wong et al. (2020) did not find any relationship between MP levels and population density [53]. Based on these results and the lack of consensus in the scientific community, the variation of the concentration and the change in composition need to be better investigated according to the potential global vs local point source. An alternative explanation could be the increase in salinity as the water flows from the fluvial to the estuarine sampling sites. Some work has shown a positive correlation between MP levels and increased salinity [18,20]. For example, a similar study found higher concentrations of MP in both surface water and bivalve tissue in the St. Lawrence estuarine and gulf as opposed to the river [20]. This highlights the importance of reporting salinity levels when conducting environmental MP monitoring or conducting lab MP fate experiments. Altogether, this work offers valuable insights into the distribution and behavior of MPs within the SLRE basin, thus contributing to the preservation and management of water resources.

## Supporting information

**S1 Table. Akaike Information Criterion on small sample size (AICc) for competing models ranked from most to least parsimonious with area, site, and mesh size as independent variables to explain the sum of microplastic particles combined and subsetted for fiber and fragments.**
(DOCX)

**S2 Table.** Shannon and Simpson diversity index for site and area using the vegan package with function "adonis" (n = 61). Permutations were set at 999.
(DOCX)

## Acknowledgments

The authors would like to thank *Stratégies Saint-Laurent* for their help on fieldwork, Poly-Mer© and Québec Océan for using their nets and Caroline Guilmette (Université Laval), JM Gutierrez-Villagomez (INRS) and AE Tremblay (INRS) for fieldwork and MP filtrations. We also thank Steve Vissault for reviewing R scripts and Scott Hepditch for proof-reading the final version of the manuscript.

## Author contributions

**Conceptualization:** Valérie S. Langlois, Tuan Anh S. To, Julien Gigault.

**Data curation:** Valérie S. Langlois, Tuan Anh S. To, Raphael A Lavoie.

**Formal analysis:** Raphael A Lavoie.

**Funding acquisition:** Valérie S. Langlois, Julien Gigault.

**Investigation:** Valérie S. Langlois, Tuan Anh S. To, Julien Gigault.

**Methodology:** Tuan Anh S. To, Eve S. Larocque.

**Project administration:** Valérie S. Langlois.

**Resources:** Valérie S. Langlois, Raphael A Lavoie.

**Supervision:** Valérie S. Langlois.

**Validation:** Valérie S. Langlois, Raphael A Lavoie.

**Visualization:** Raphael A Lavoie.

**Writing – original draft:** Valérie S. Langlois.

**Writing – review & editing:** Valérie S. Langlois, Tuan Anh S. To, Julien Gigault, Raphael A Lavoie.

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
