## [Decision Letter · Decision Letter 0]

18 Sep 2024

PONE-D-24-18181Surface Water Microplastics in the St. Lawrence River and Estuary in CanadaPLOS ONE

Dear Dr. Langlois, 

Thank you for submitting your manuscript to PLOS ONE. After careful consideration, we feel that it has merit but does not fully meet PLOS ONE’s publication criteria as it currently stands. Therefore, we invite you to submit a revised version of the manuscript that addresses the points raised during the review process.

We look forward to receiving your revised manuscript.

Kind regards,

Beizhan Yan

Academic Editor

PLOS ONE

Journal Requirements:

3. Please note that PLOS ONE has specific guidelines on code sharing for submissions in which author-generated code underpins the findings in the manuscript. In these cases, all author-generated code must be made available without restrictions upon publication of the work.

Please review our guidelines at https://journals.plos.org/plosone/s/materials-and-software-sharing#loc-sharing-code and ensure that your code is shared in a way that follows best practice and facilitates reproducibility and reuse.

5. Please expand the acronym “CNRS” (as indicated in your financial disclosure) so that it states the name of your funders in full.

6. Thank you for stating the following financial disclosure:

“The authors acknowledge the funding from the Canada Research Chair Program to VSL and from the CNRS to JG.”

Please state what role the funders took in the study. If the funders had no role, please state: "The funders had no role in study design, data collection and analysis, decision to publish, or preparation of the manuscript". If this statement is not correct you must amend it as needed.

7. Please note that funding information should not appear in the Acknowledgments section or other areas of your manuscript. We will only publish funding information present in the Funding Statement section of the online submission form. Please remove any funding-related text from the manuscript.

8. In the online submission form, you indicated that “All data files are available upon request.”.

1) In a public repository,

2) Within the manuscript itself, or

3) Uploaded as supplementary information.

9. We note that Figure 1 in your submission contain map images which may be copyrighted. All PLOS content is published under the Creative Commons Attribution License (CC BY 4.0), which means that the manuscript, images, and Supporting Information files will be freely available online, and any third party is permitted to access, download, copy, distribute, and use these materials in any way, even commercially, with proper attribution. For these reasons, we cannot publish previously copyrighted maps or satellite images created using proprietary data, such as Google software (Google Maps, Street View, and Earth). For more information, see our copyright guidelines: http://journals.plos.org/plosone/s/licenses-and-copyright.

1) You may seek permission from the original copyright holder of Figure 1 to publish the content specifically under the CC BY 4.0 license. 

2) If you are unable to obtain permission from the original copyright holder to publish these figures under the CC BY 4.0 license or if the copyright holder’s requirements are incompatible with the CC BY 4.0 license, please either i) remove the figure or ii) supply a replacement figure that complies with the CC BY 4.0 license. Please check copyright information on all replacement figures and update the figure caption with source information. If applicable, please specify in the figure caption text when a figure is similar but not identical to the original image and is therefore for illustrative purposes only.

10. Please include captions for your Supporting Information files at the end of your manuscript, and update any in-text citations to match accordingly. Please see our Supporting Information guidelines for more information: http://journals.plos.org/plosone/s/supporting-information.

Reviewers' comments:

Reviewer's Responses to Questions

**Comments to the Author**

1. Is the manuscript technically sound, and do the data support the conclusions?

Reviewer #1: Partly

Reviewer #2: Yes

2. Has the statistical analysis been performed appropriately and rigorously? 

Reviewer #1: Yes

Reviewer #2: Yes

3. Have the authors made all data underlying the findings in their manuscript fully available?

Reviewer #1: No

Reviewer #2: Yes

4. Is the manuscript presented in an intelligible fashion and written in standard English?

Reviewer #1: Yes

Reviewer #2: Yes

5. Review Comments to the Author

Reviewer #1: Conclusion is supported by the results and methodological approach. Statistical analysis is sound. Manuscript is understandable, but more details need to be added throughout. All data need to be available at the time of manuscript submission, not available upon request. Please provide all underlying datasets used to inform this study.

Reviewer #2: The manuscript "Identification of the Driving Factors of Microplastic Load and Morphology in Estuaries for Improving Monitoring and Management Strategies: A Global Meta-analysis" was well organized and provided a comprehensive understanding of the morphological characteristics and polymers of microplastics in the St. Lawrence River and Estuary in Canada. However, there are still some concerns as follows, which did not be clarified in the manuscript. The English language of the manuscript needs to be improved by a native English speaker, particular in there are too many long sentences without commas in the manuscript. Overall, I recommend to publish the manuscript after the minor revisions.

Specific comments

L33-34: Too specific, it should be more macroscopic, the harmful of microplastics are not limited on living organisms.

L37: Confirm if the use of commas is correct.

L41: Please use passive tense while presenting the results. Do not use "our".

L41: It is best to indicate the proportion

L45-46: Please rewrite the sentence.

L50: Please indicate the definition of microplastics when it first appears.

L79-81: Change to “Firstly, secondly”.

L120: How many (proportion) Potential MPs were detected by ATR-FTIR.

L162: First describe the shape, then talk about the polymer, and what about the concentration?

L203: Please bring appropriate references regarding your statement.

L218: Plastics or microplastics; in China or in Chinese river?

L228: In a previous study

L235: Please rewrite the sentence.

Figure 2 lacks a scale bar

Figure 5 can add color to make it look better

6. PLOS authors have the option to publish the peer review history of their article (what does this mean? ). If published, this will include your full peer review and any attached files.

**Do you want your identity to be public for this peer review?** For information about this choice, including consent withdrawal, please see our Privacy Policy .

Reviewer #1: No

Reviewer #2: No

---

## [Author Response · Author response to Decision Letter 1]

27 Sep 2024

Editor’s comments:

Done.

Done. This sentence was added. No permits were required to sample water in the SLRE.

3. Please note that PLOS ONE has specific guidelines on code sharing for submissions in which author-generated code underpins the findings in the manuscript. In these cases, all author-generated code must be made available without restrictions upon publication of the work.

Please review our guidelines at https://journals.plos.org/plosone/s/materials-and-software-sharing#loc-sharing-code and ensure that your code is shared in a way that follows best practice and facilitates reproducibility and reuse.

Done. The code is shared on an open access repository on GitHub at https://github.com/ECCC-lavoie-ecotox/MPs_St-Lawrence in a way that follows best practice and facilitates reproducibility and reuse. This was added at the end of the Methodology section. As we needed helped for this, we have added this line to our Acknowledgements: We thank Steve Vissault for reviewing R scripts.

Done.

5. Please expand the acronym “CNRS” (as indicated in your financial disclosure) so that it states the name of your funders in full.

Done. CNRS: Centre national de la recherche scientifique. This got added to the cover letter.

6. Thank you for stating the following financial disclosure:

“The authors acknowledge the funding from the Canada Research Chair Program to VSL and from the CNRS to JG.”

Please state what role the funders took in the study. If the funders had no role, please state: "The funders had no role in study design, data collection and analysis, decision to publish, or preparation of the manuscript". If this statement is not correct you must amend it as needed.

Done. We have added your suggested statement to the cover letter as requested.

7. Please note that funding information should not appear in the Acknowledgments section or other areas of your manuscript. We will only publish funding information present in the Funding Statement section of the online submission form. Please remove any funding-related text from the manuscript.

We have ensured to not mention funding in the Acknowledgments.

8. In the online submission form, you indicated that “All data files are available upon request.”.

1) In a public repository,

2) Within the manuscript itself, or

3) Uploaded as supplementary information.

We have attached the raw data to this revised submission. Please change the data availability box for us.

9. We note that Figure 1 in your submission contain map images which may be copyrighted. All PLOS content is published under the Creative Commons Attribution License (CC BY 4.0), which means that the manuscript, images, and Supporting Information files will be freely available online, and any third party is permitted to access, download, copy, distribute, and use these materials in any way, even commercially, with proper attribution. For these reasons, we cannot publish previously copyrighted maps or satellite images created using proprietary data, such as Google software (Google Maps, Street View, and Earth). For more information, see our copyright guidelines: http://journals.plos.org/plosone/s/licenses-and-copyright.

1) You may seek permission from the original copyright holder of Figure 1 to publish the content specifically under the CC BY 4.0 license.

2) If you are unable to obtain permission from the original copyright holder to publish these figures under the CC BY 4.0 license or if the copyright holder’s requirements are incompatible with the CC BY 4.0 license, please either i) remove the figure or ii) supply a replacement figure that complies with the CC BY 4.0 license. Please check copyright information on all replacement figures and update the figure caption with source information. If applicable, please specify in the figure caption text when a figure is similar but not identical to the original image and is therefore for illustrative purposes only.

Shapefiles were taken from Open Canada, which is compatible with the CC BY 4.0 license. The following statement was included in the caption of Figure 1: Basemap source: Government of Canada; Natural Resources Canada (https://open.canada.ca/data/en/dataset/8ba2aa2a-7bb9-4448-b4d7-f164409fe056) and Natural Earth (public domain): http://www.naturalearthdata.com/. Contains information licensed under the Open Government License – Canada (https://open.canada.ca/en/open-government-licence-canada) and Natural Earth (https://www.naturalearthdata.com/about/terms-of-use/).

10. Please include captions for your Supporting Information files at the end of your manuscript, and update any in-text citations to match accordingly. Please see our Supporting Information guidelines for more information: http://journals.plos.org/plosone/s/supporting-information.

Data are shared on an open access repository on GitHub at https://github.com/ECCC-lavoie-ecotox/MPs_St-Lawrence in a way that follows best practice and facilitates reproducibility and reuse. This was added at the end of the Methodology section.

Reviewer #1: Conclusion is supported by the results and methodological approach. Statistical analysis is sound. Manuscript is understandable, but more details need to be added throughout. All data need to be available at the time of manuscript submission, not available upon request. Please provide all underlying datasets used to inform this study.

Thanks for the positive comments. Dataset were added to the submission.

1) Summary of manuscript and overall impression:

This study quantified and characterized microplastics in the surface waters of the St. Lawrence River and Estuary (SLRE) using nets with two different mesh sizes, to fill a gap in our understanding of microplastic abundance in the SLRE’s bulk water. The SLRE has been sampled previously for microplastic, but the novelty of this study is that it investigates the relationship between microplastic abundance and salinity. Authors predicted that microplastic concentrations will scale with salinity and urbanness of land use. Microplastics were detected at every site, concentrations of fragments were higher downstream than upstream, and there is some evidence that assemblage structure of MPs varied between fluvial and estuarine portions of the SLRE system, supporting the prediction that salinity plays a role in MP fate.

The question of the relationship between microplastic concentration and salinity is interesting and novel. The statistics and figures presented are also interesting. Most importantly, the science is sound, which is the criterion for publication in PLOS ONE. Recommendation: major revision. However, I recommend the writing be clearer throughout the manuscript, and also authors take the time to explain more details in the methods, results, and discussion sections. There is no word limit, and there are parts of the manuscript that can be flushed out more. Also at this moment, there are methodological details that are missing and preclude replication of this study. In addition, all data need to be available at the time of manuscript submission, not available upon request. See comments below.

Thanks for the positive comments. Dataset were added to the submission.

1) Questions and comments:

How were blanks collected? E.g. distilled water poured onto net, then resuspended in the same way as samples? How much distilled water was poured onto net for each blank?

A 1-L distilled water was poured into a glass on the boat and put at 4 oC until filtration. This has been specified in the text.

Authors say that several digestion methods were tested; please list these methods somewhere in the manuscript or SI. Also remember to refer to the raw data in your repository.

We have tested the digestion with hydroperoxide, H2O2 30 % (v/v) for different times and temperatures: 6 h at 65 oC, 24 h at 50 oC, 24 h at 25 oC, and 4 h at 50 oC with Fe(II).

This was adjusted in the text.

Fiber concentrations inferred using Manta net sampling is not accurate, because Manta nets underestimate particles smaller than its net mesh. For fibers, its diameter is typically 10-20 um, meaning it can slip through the manta net. Please add a line in your manuscript about how your fiber concentrations in this river system should be interpreted with caution because Manta nets are not designed to measure fibers. Hung et al. (2020): “Capture of these particles is likely by random chance and is expected to be an underestimate of the total number of these small‐sized particles. For this reason, the quantities of particles with dimension smaller than the mesh size should be treated with caution and considered qualitative estimates.” https://doi.org/10.1002/ieam.4325 Also see Zhu et al. (2021): “Two surface water collection methods were employed: manta trawl (22,28) with a 333 μm mesh and 1 L grab. Manta samples (N = 65) were collected to quantify and characterize all microdebris, except for microfibers, which can slip through the holes in the mesh due to their shape.” https://pubs.acs.org/doi/full/10.1021/acsestwater.0c00292

Thanks for pointing this out. We have added your suggested comments to the text and use the Zhu et al. citation. Caution should also be considered when interpreting fiber MPs caught by the Manta as its underestimate particles smaller than its net mesh [45]. Therefore, the used of smaller mesh size nets like PolyMer (this study) is favored to improve fiber MP estimations.

For results section, please add this basic information: average +/- SD for all 11 sites combined, which site had highest MP concentration (average +/- SD), which site had lowest MP concentration (average +/- SD). I’m not sure why you are using this new measure of concentration, “particles of plastic per million liters” or PPML; instead of inventing a new unit, please consider using existing units to make your study more comparable to others. See Bucci et al. (2020): usually, units of concentration used in ecotoxicology tests are “per mL” or “per L”. More and more often, ecotoxicologists are trying to use environmentally relevant concentrations in their experiments. To make their lives easier, and to harmonize reporting across studies, please consider reporting in a unit that other people use. Unless you provide justification for why you are using a new unit. Please report the average +/- SD concentration for your blanks. Did you blank correct?

The information on average +/- SD for all 11 sites combined, which site had highest MP concentration (average +/- SD), which site had lowest MP concentration (average +/- SD) was added in the text: MP concentrations (mean +/- SD) for all 11 sites combined was 72.3 ± 93.0 PPML and was highest at the La Pocatière site (111.0 ± 120.0 PPML) and lowest at the Sorel site (38.7 ± 40.8 PPML).

This unit (PPML) was simply used because of the small number of MPs in the samples. Otherwise, the numbers would have been too long to report (too many 0.0000). The average +/- SD were added. Yes, we blank corrected.

For methods section: Equation 1 implies that you only counted half the particles on your filter. State this explicitly somewhere in the Methods section that you subsampled. Why – to save time? Because the filters had a lot of particles on them? Please provide justification.

We kept the second halve of the filters to study nanoplastics, which will be part of another project. We have clarified this in the text.

How many particles (%) were selected for FTIR analysis? See de Frond et al. (2022) for minimum number of particles that need to be identified in microplastic samples: https://www.sciencedirect.com/science/article/abs/pii/S0045653522032659?via%3Dihub

All particles were analyzed by FTIR.

Discussion: please discuss potential limitations of this study

We have added the Manta net size limit in catching fibers to the discussion.

Line by line:

Equation 1: does ‘debit’ mean ‘volume’?

Yes. Sorry for this translation issue. It was fixed.

Figure 1: the white is supposed to indicate water, but the US is also shown in white. Please consider making the US a different colour to indicate land.

Water is now blue and land is white.

Figure 3: colours are a bit harsh to the eye, and also they are not colourblind-friendly. Please consider using a different colour palette, e.g. viridis. I verified this using a chromatic vision simulator app – you can download this app yourself and test the colour palette you end up choosing.

We changed the colour palette, so they are colourblind-friendly.

Figure 5 caption: “areas are separated by a vertical dashed line”. What areas? fluvial vs estuary? Please add this information to your figure caption; make sure captions in general are more informative.

Done.

Line 35: “we a

---

## [Decision Letter · Decision Letter 1]

25 Nov 2024

PONE-D-24-18181R1Surface Water Microplastics in the St. Lawrence River and Estuary in CanadaPLOS ONE

Dear Dr. Langlois,

Thank you for submitting your manuscript to PLOS ONE. After careful consideration, we feel that it has merit but does not fully meet PLOS ONE’s publication criteria as it currently stands. Therefore, we invite you to submit a revised version of the manuscript that addresses the points raised during the review process.

We look forward to receiving your revised manuscript.

Kind regards,

Phuping Sucharitakul

Academic Editor

PLOS ONE

Additional Editor Comments:

Dear Authors,

The reviewers have submitted their comments and recommendations. Please carefully revise the manuscript, ensuring you address each comment thoroughly and provide well-reasoned responses. Since one of the reviewers has recommended rejecting the manuscript, it is crucial to include a strong rebuttal with clear and compelling reasons to support why the manuscript should be accepted both in the cover letter and response to reviewers files.

Regards,

Phuping Sucharitakul

Reviewers' comments:

Reviewer's Responses to Questions

**Comments to the Author**

1. If the authors have adequately addressed your comments raised in a previous round of review and you feel that this manuscript is now acceptable for publication, you may indicate that here to bypass the “Comments to the Author” section, enter your conflict of interest statement in the “Confidential to Editor” section, and submit your "Accept" recommendation.

Reviewer #1: (No Response)

Reviewer #2: All comments have been addressed

Reviewer #3: (No Response)

2. Is the manuscript technically sound, and do the data support the conclusions?

Reviewer #1: Yes

Reviewer #2: Yes

Reviewer #3: (No Response)

3. Has the statistical analysis been performed appropriately and rigorously? 

Reviewer #1: I Don't Know

Reviewer #2: Yes

Reviewer #3: (No Response)

4. Have the authors made all data underlying the findings in their manuscript fully available?

Reviewer #1: Yes

Reviewer #2: Yes

Reviewer #3: (No Response)

5. Is the manuscript presented in an intelligible fashion and written in standard English?

Reviewer #1: Yes

Reviewer #2: Yes

Reviewer #3: (No Response)

6. Review Comments to the Author

Reviewer #1: 1. Reviewer: How were blanks collected? E.g. distilled water poured onto net, then resuspended in the same way as samples? How much distilled water was poured onto net for each blank?

Response: A 1-L distilled water was poured into a glass on the boat and put at 4 oC until filtration.

This has been specified in the text.

Reviewer: You should always refer to the specific line numbers where the changes were made. This will make your reviewers’ lives a lot easier.

2. Reviewer: Authors say that several digestion methods were tested; please list these methods somewhere in the manuscript or SI. Also remember to refer to the raw data in your repository.

Response: We have tested the digestion with hydroperoxide, H2O2 30 % (v/v) for different times

and temperatures: 6 h at 65 oC, 24 h at 50 oC, 24 h at 25 oC, and 4 h at 50 oC with

Fe(II).

This was adjusted in the text.

Reviewer: Line 132: “hydroperoxide” should be “hydrogen peroxide” (See https://en.wikipedia.org/wiki/Hydroperoxide).

3. Reviewer: Fiber concentrations inferred using Manta net sampling is not accurate, because Manta nets underestimate particles smaller than its net mesh. For fibers, its diameter is typically 10-20 um, meaning it can slip through the manta net. Please add a line in your manuscript about how your fiber concentrations in this river system should be interpreted with caution because Manta nets are not designed to measure fibers.

Hung et al. (2020): “Capture of these particles is likely by random chance and is expected to be an underestimate of the total number of these small‐sized particles. For this reason, the quantities of particles with dimension smaller than the mesh size should be treated with caution and considered qualitative estimates.” https://doi.org/10.1002/ieam.4325

Also see Zhu et al. (2021): “Two surface water collection methods were employed: manta trawl (22,28) with a 333 μm mesh and 1 L grab. Manta samples (N = 65) were collected to quantify and characterize all microdebris, except for microfibers, which can slip through the holes in the mesh due to their shape.” https://pubs.acs.org/doi/full/10.1021/acsestwater.0c00292

Response: Thanks for pointing this out. We have added your suggested comments to the text and

use the Zhu et al. citation. Caution should also be considered when interpreting fiber

MPs caught by the Manta as its underestimate particles smaller than its net mesh [45].

Therefore, the used of smaller mesh size nets like PolyMer (this study) is favored to

improve fiber MP estimations.

Reviewer: Microfiber diameters tend to range between 10-20 um. They would still slip through your PolyMer net. I would remove your second sentence because it is not justified.

4. Reviewer: For results section, please add this basic information: average +/- SD for all 11 sites combined, which site had highest MP concentration (average +/- SD), which site had lowest MP concentration (average +/- SD). I’m not sure why you are using this new measure of concentration, “particles of plastic per million liters” or PPML; instead of inventing a new unit, please consider using existing units to make your study more comparable to others. See Bucci et al. (2020): usually, units of concentration used in ecotoxicology tests are “per mL” or “per L”. More and more often, ecotoxicologists are trying to use environmentally relevant concentrations in their experiments. To make their lives easier, and to harmonize reporting across studies, please consider reporting in a unit that other people use. Unless you provide justification for why you are using a new unit. Please report the average +/- SD concentration for your blanks. Did you blank correct?

Response: The information on average +/- SD for all 11 sites combined, which site had highest

MP concentration (average +/- SD), which site had lowest MP concentration (average

+/- SD) was added in the text: MP concentrations (mean +/- SD) for all 11 sites

combined was 72.3 ± 93.0 PPML and was highest at the La Pocatière site (111.0 ±

120.0 PPML) and lowest at the Sorel site (38.7 ± 40.8 PPML).

This unit (PPML) was simply used because of the small number of MPs in the samples.

Otherwise, the numbers would have been too long to report (too many 0.0000). The

average +/- SD were added. Yes, we blank corrected.

Reviewer: The PPML still kind of bothers me, but I understand where the authors are coming from.

5. Reviewer: Lines 233-234: you say concentrations never exceed 0.001 particles per L – for surface water? And where – in US and Canada? Please make your writing a bit clearer, more informative. But in Rowenczyk et al. (2022), they found a maximum of 0.032 MPs per L in SLRE middle portion – contradiction? Also in Rowenczyk et al., they use particles/L. In order to make your results easily comparable to theirs, doesn’t it make sense to use the same units? I have not seen PPML used before in the literature. Two possibilities: 1) PPML has not been used before, then consider not creating a new unit; we already have a harmonization problem in the plastics field. 2) PPML has been used before: even if this is the case, it deviates from the norm and also makes harmonization difficult.

Response: Thanks for pointing this out, we have revamped the paragraph as: We expected to find

higher MP concentrations in surface waters upstream than downstream considering

the high population density living nearby the Great Lakes and SLRE from the United

States and Canada, but lower concentrations were found than elsewhere. For

example, Luo et al. (2019) measured concentrations around 2,000,000 PPML near

major urban centers in China [46]. This difference can be explained by the hydrology and typology of the SLRE. Indeed, studies have shown that shoreline accumulation

and deposition in the sediment were not considered as they can be collected and

decontaminated [22]. Castañeda et al. (2014) have estimated MP loadings in SLRE

sediment up to 3,800,000 PPML [22] at sites which have accumulated continuously for

over 50 years compared to transitioning surface water [47]. In another study, fragments

and fibers are the most prevalent particles found in the sediment [48] in South Africa,

which could correlate with the variation in shape observed from the sampling

campaign.

The second part of the question was addressed above.

Reviewer: If you start your sentence with “We expected to find

higher MP concentrations in surface waters upstream than downstream considering

the high population density living nearby the Great Lakes and SLRE from the United

States and Canada…” then I would expect you to end the sentence discussing what you actually found regarding the differences between upstream and downstream MP concentrations. Instead, you talk about an unconnected topic, which is how your concentrations are lower compared to other locations. Consider splitting into two separate sentences.

Comparing water to sediment doesn’t make sense – it’s like comparing apples to oranges.

This entire paragraph consists of a series of sentences with unrelated themes strung together. First you talk about surface water. Then you talk about sediment. Then you added a sentence about MP morphology, which has no connection to the previous sentences. I suggest separating the sentences to form their own paragraphs, so that each paragraph begins with a topic sentence and the rest of the paragraph expands on that topic sentence.

***

I do not need to see the manuscript again. If all of the above changes are made, then I recommend this manuscript for publication.

Reviewer #2: (No Response)

Reviewer #3: General Comment:

In the manuscript “PONE-D-24-18181R1,” the authors present a study on the presence and behavior of microplastics (MPs) on the water surface at 11 sites along the St. Lawrence River and Estuary in Canada. Using nets with different mesh sizes (100 µm and 300 µm) for collection and FTIR analysis, the authors identified MPs predominantly composed of fibers, fragments, and spheres, with polyester, polyethylene, and polypropylene being the most common polymers. The study also explores the relationship between salinity gradients and the aggregation of MPs, emphasizing the importance of this factor in estuarine systems. While the topic of the manuscript is relevant given the growing interest in MPs in estuarine environments and their potential environmental impacts, the study has significant deficiencies that justify a recommendation for rejection in a high-impact journal like PLOS ONE.

Specific Comments:

1. The scientific contribution is primarily incremental and lacks substantial innovation. The relationship between salinity and MP aggregation, which is a central focus of the study, has already been addressed in existing literature. The manuscript does not present a theoretical or experimental approach that meaningfully challenges or expands current understanding.

2. The study’s regional focus on the St. Lawrence River and Estuary limits its global applicability and reduces its appeal to a diverse international audience, which is the primary readership of PLOS ONE.

3. The introduction of the unit "particles per million liters" (PPML) is not convincingly justified and offers no clear advantage over widely accepted standardized units, such as particles per liter. This choice hampers comparability with global studies and compromises the harmonization of results within the field.

4. While the use of nets with different mesh sizes is methodologically interesting, it does not represent a significant advancement in the field. Instead, it highlights known limitations of sampling methods without proposing creative solutions or methodological innovations that advance the discipline. For a journal like PLOS ONE, studies are expected to contribute significantly to the progression of scientific knowledge through novel theories, innovative methodologies, or results with broad applicability.

5. Although the authors mention anti-contamination measures, their description is superficial and insufficient to inspire full confidence in the results. Essential details, such as whether laminar flow chambers were used during sample handling, the type of clothing worn by operators to prevent contamination with synthetic fibers (e.g., cotton or disposable lab coats), and the frequency of contamination controls in the laboratory, are not adequately discussed. Moreover, while field blanks are noted as a good practice, there is no detailed explanation of how these control values were used to adjust the reported concentrations of MPs in environmental samples.

6. The lack of comprehensive details about anti-contamination procedures undermines the methodological robustness of the study. Given the ubiquity of plastics in laboratory and field environments, studies on MPs require stringent safeguards against external contamination. Although the authors acknowledge the importance of preventing contamination, the superficiality of their descriptions compromises the transparency and reliability of their data, thereby calling into question the validity of their conclusions.

7. Thus, although the manuscript demonstrates technical merit, it fails to meet the criteria for theoretical impact and innovation expected for publication in a high-impact journal. I recommend that the authors consider submitting their study to a regional or specialized journal, where its incremental contributions may be better appreciated and appropriately valued.

7. PLOS authors have the option to publish the peer review history of their article (what does this mean? ). If published, this will include your full peer review and any attached files.

**Do you want your identity to be public for this peer review?** For information about this choice, including consent withdrawal, please see our Privacy Policy .

Reviewer #1: No

Reviewer #2: No

Reviewer #3: No

---

## [Author Response · Author response to Decision Letter 2]

25 Nov 2024

We have addressed all the reviewers’ minor comments in the attached document. It is unfortunate to get a third reviewer seven months after first submission to PLOS ONE. That reviewer who stays anonymous is so negative. It has been months that the authors are redoing statistical analyses, graphs, maps, etc. to please the first two reviewers. The first two reviewers and the authors disagree with this new reviewer: the topic is timely and relevant to the readers of PLOS ONE; plus, most of the new reviewer’s comments were already discussed by the previous reviewers. Therefore, we urge the Editor to disregard this new reviewer’s review as the manuscript has new satisfied the main reviewers.

---

## [Editor Report · Decision Letter 2]

1 Dec 2024

Surface Water Microplastics in the St. Lawrence River and Estuary in Canada

PONE-D-24-18181R2

Dear Dr. Langlois,

We’re pleased to inform you that your manuscript has been judged scientifically suitable for publication and will be formally accepted for publication once it meets all outstanding technical requirements.

Kind regards,

Phuping Sucharitakul

Academic Editor

PLOS ONE
---

## [Editor Report · Acceptance letter]

PONE-D-24-18181R2

PLOS ONE

Dear Dr. Langlois,

I'm pleased to inform you that your manuscript has been deemed suitable for publication in PLOS ONE. Congratulations! Your manuscript is now being handed over to our production team.

Kind regards,

on behalf of

Dr. Phuping Sucharitakul

Academic Editor

PLOS ONE